# Was the risk of death among the population of teachers and other school workers in England and Wales due to COVID-19 and all causes higher than other occupations during the pandemic in 2020? An ecological study using routinely collected data on deaths from the Office for National Statistics

Sarah J Lewis ![ORCID],[1,2] Kyle Dack,[1,2] Caroline L Relton,[1,2] Marcus R Munafo,[2,3] George Davey Smith ![ORCID] [1,2]

For numbered affiliations see end of article.

**Correspondence to**
Professor Sarah J Lewis;
S.J.Lewis@bristol.ac.uk

### ABSTRACT

**Objectives** To estimate occupation risk from COVID-19 among teachers and others working in schools using publicly available data on mortality in England and Wales.

**Design** Analysis of national death registration data from the Office for National Statistics.

**Setting** England and Wales, 8 March–28 December 2020, during the COVID-19 pandemic.

**Participants** The total working age population in England and Wales plus those still working aged over 65 years.

**Primary and secondary outcomes** Death with COVID-19 as a primary outcome and death from all causes as a secondary outcome.

**Results** Across occupational groups, there was a strong correlation between COVID-19 mortality and both non-COVID-19 and all-cause mortality. The absolute mortality rates for deaths with COVID-19 were low among those working in schools (from 10 per 100 000 in female primary school teachers to 39 per 100 000 male secondary school teachers) relative to many other occupations (range: 9–50 per 100 000 in women; 10–143 per 100 000 in men). There was weak evidence that secondary school teachers had slightly higher risks of dying with COVID-19 compared with the average for all working-aged people, but stronger evidence of a higher risk compared with the average for all professionals; primary school teachers had a lower risk. All-cause mortality was also higher among all teachers compared with all professionals. Teaching and lunchtime assistants were not at higher risk of death from COVID-19 compared with all working-aged people.

**Conclusion** There was weak evidence that COVID-19 mortality risk for secondary school teachers was above expectation, but in general school staff had COVID-19 mortality risks which were proportionate to their non-COVID-19 mortality risk.

### Strengths and limitations of this study

► We used routinely collected data on all deaths in England and Wales, which are near complete and not susceptible to serious ascertainment biases.

► We were able to compare mortality data for teachers and other school workers with all other occupational groups and with the working-aged population.

► The number of deaths due to COVID-19 was small and thus differences between the specific occupational groups were imprecisely estimated.

► We did not have access to individual-level mortality data so were not able to account for potential confounders such as comorbidities or household size.

► For those working in schools who were aged over 65 years, we had neither mortality rates per 100 000 nor total numbers within the group; we only had number of deaths and a 5-year average, we do not know whether the denominators have changed for this group over the last 5 years.

### BACKGROUND

School closures have been implemented in many countries in an effort to slow the spread of SARS-CoV-2 responsible for the COVID-19 pandemic. This has affected hundreds of millions of children globally and has been contentious due to harms that have arisen to children as a result of school closures.[1] Schools in England and Wales closed to most pupils (except a small number of children of critical workers and some vulnerable children) on 20 March 2020 and did not reopen fully until September 2020; they closed

again for the Christmas holidays (between 11 and 18 December) and reopened on 8 March 2021 in England. In Wales, the reopening of schools after Christmas was more gradual and started on 22 February 2021 with the youngest children, with all children returning by 15 April 2021.

Whether or not teachers and others working in schools are at higher risk of COVID-19 as a result of schools being open is central to decisions on school closures, but until recently there has not been good data on this. Record linkage studies carried out in Scotland[2] and Finland[3] show that teachers had a greater risk of being diagnosed as a COVID-19 case compared with the general population when schools were open; there was a 40% higher risk of COVID-19 among teachers in the Scottish study, and between 50% and 70% higher risk in the Finnish study. A study in Sweden found that teachers delivering in-person teaching had a twofold higher risk of COVID-19 infection compared with those who were teaching remotely.[4] However, studies examining transmission in schools have found this to occur at low levels.[5 6] One reason for the discrepancy is that teachers are likely to be under greater surveillance, and therefore undergo more testing than those working in other occupations. In the Scottish study, teachers were much more likely to be tested over the autumn term compared with the general population, and this is likely to explain at least some of the elevated number of cases among teachers.

Analyses of more robust outcomes such as hospitalisation from COVID-19 are less susceptible to detection bias and may be more reliable. The Scottish study[2] found teachers and their household members were at a lower risk of being hospitalised with COVID-19 compared with adults of working age in the general population after matched on age, sex and general practice registration and adjusted for ethnicity, deprivation, comorbidity counts and sharing a household with a healthcare worker. Sweden has been an outlier in the COVID-19 pandemic in that schools have remained open for children up to 16 years of age. A study by the Swedish Public Health Agency found that 20 out of 103 596 school teachers in Sweden received intensive care treatment up to 30 June 2020; an age-matched and sex-matched analysis comparing teachers with other occupations (except healthcare workers) reported an OR of 0.43 (95% CI 0.28 to 0.68).[7] Similarly, a study carried out in Norway did not find higher rates of hospitalisation with COVID-19 for school teachers, after adjustment for age, sex and country of birth.[8] There is an absence of evidence on the risk to teaching and lunchtime assistants working in schools.

The Office for National Statistics (ONS) routinely collects mortality data for those living in England and Wales. They released data on deaths with COVID-19 and all-cause mortality by occupation as a publicly available dataset.[9] We used these data to compare mortality risk among school workers and all working-aged adults in England and Wales. We also compared mortality rates among teachers with all professionals, because there are large differences in mortality risk by occupation, largely driven by differences in socioeconomic circumstances.[10]

## METHODS
The ONS mortality statistics used for this analysis are based on mandatory registration of all deaths in England and Wales. So far, three datasets on mortality with COVID-19 by occupation have been released; the first covered the period from 8 March to 20 April which included the peak of wave 1 deaths, the second covered the period between 8 March and 25 May, and the final dataset (the one we used for the analysis in this paper) covered the period from 8 March to 28 December.

We used these data to: (1) describe the mortality rates among all educational professionals combined and present these alongside rates for all professionals and all working-aged people for the different time periods covered by these data (table 1); (2) compare the number of deaths with historical data on deaths for the same occupational group over 5 years (2015–2019) for all school workers and for individual occupations (teachers, teaching assistants, school secretaries, etc), which are presented alongside death rates among all occupations and all professional occupations over the same period of the pandemic (table 2 and online supplemental table S1); (3) calculate the risk of COVID-19 and all-cause mortality among teachers and other school staff compared with all working-aged people (tables 3 and 4) and all professionals (online supplemental tables S2 and S3); and finally, (4) we investigated the ratio of mortality with COVID-19 to mortality from other causes among different occupational groups to determine whether school staff were outliers (online supplemental tables S4–S6 and all figures).

### Occupational exposure groups
Occupation was reported on the death certificate at the time of death registration by the informant and was coded according to the ONS standard occupational classification 2010.[11] Population counts for occupational groups were estimated from the Annual Population Survey (APS), conducted in 2019.[12] The APS is the largest ongoing household survey in the UK, conducting interviews with members of randomly selected households.

The group 'teaching and educational professionals', as defined by the ONS, includes: higher education teaching professionals, further education teaching professionals, secondary education teaching professionals, primary and nursery education teaching professionals, special needs education teaching professionals, senior professionals of educational establishments, education advisers and school inspectors, and teaching and other educational professionals not elsewhere defined.

The group 'teaching and educational professionals' does not include: teaching assistants, school secretaries, lunchtime assistants or educational support assistants, but these occupations were considered along with teachers working in schools where sufficient data were available.

**Table 1** Age-adjusted all-cause and with COVID-19 mortality rates per 100 000 population (95% CIs) for all educational professionals, all professional occupations and all working-aged adults in England and Wales by overlapping time periods covered by these data

| Occupational group | Cause of mortality | Men | | | Women | | |
|---|---|---|---|---|---|---|---|
| | | 8 March–20 April | 8 March–25 May | 8 March–28 December | 8 March–20 April | 8 March–25 May | 8 March–28 December |
| Working-aged adults | With COVID-19 | 9.9 (9.4 to 10.4) | 19.1 (18.4 to 19.8) | 31.4 (30.6 to 32.3) | 5.2 (4.9 to 5.6) | 9.7 (9.3 to 10.2) | 16.8 (16.2 to 17.5) |
| | All cause | 41.6 (40.6 to 42.6) | 78.1 (76.7 to 79.4) | 256.0 (253.5 to 258.4) | 26.3 (25.5 to 27.1) | 48.4 (47.3 to 49.4) | 158.3 (156.4 to 160.2) |
| | Ratio* | 1:4.2 | 1:4.1 | 1:8.2 | 1:5.1 | 1:5.0 | 1:9.4 |
| Professional occupations | With COVID-19 | 5.6 (4.6 to 6.6) | 11.6 (10.2 to 13.0) | 17.6 (15.9 to 19.3) | 4.2 (3.3 to 5.2) | 8.0 (6.8 to 9.3) | 12.8 (11.2 to 14.4) |
| | All cause | 22.2 (20.2 to 24.1) | 41.2 (38.5 to 43.8) | 130.4 (125.7 to 135.2) | 22.0 (19.9 to 24.1) | 39.3 (36.5 to 42.1) | 120.0 (115.2 to 124.9) |
| | Ratio* | 1:4 | 1:3.6 | 1:7.4 | 1:5.2 | 1:4.9 | 1:9.4 |
| Educational professionals | With COVID-19 | 6.7 (4.1 to 10.3) | 12.9 (9.3 to 17.4) | 18.4 (14.0 to 23.6) | 3.3 (2.0 to 4.9) | 6.0 (4.2 to 8.1) | 9.8 (7.5 to 12.5) |
| | All cause | 28.0 (22.4 to 34.5) | 48.1 (40.4 to 55.7) | 153.4 (139.8 to 167.1) | 20.4 (16.9 to 24.0) | 37.2 (32.5 to 42.0) | 110.0 (101.9 to 118.2) |
| | Ratio* | 1:4.2 | 1:3.7 | 1:8.3 | 1:6.2 | 1:6.2 | 1:11.2 |

Age-adjusted all-cause and with COVID-19 mortality per 100 000 population (95% CIs) taken from three ONS datasets on COVID-19 mortality by occupation covering different time periods during the 2020 COVID-19 pandemic. Includes the ratio of COVID-19 deaths to all-cause deaths for educational professionals, all professionals and all working-aged people (aged 20–64 years) stratified by sex.
*Ratio of deaths involving COVID-19 deaths to all deaths
ONS, Office for National Statistics.

The group defined by ONS as midday and crossing patrol occupations includes some individuals who work as road crossing patrols, however not all schools have these and if they do usually only have one, so the group will mainly comprise of those who supervise children in school during their lunch break and who prepare school lunch. Therefore, midday and crossing patrol occupations will be shorted and referred to in this manuscript as lunchtime assistants.

The comparison groups we used were:
1. All working-aged people—for comparisons of mortality rates and risk presented in tables 1, 3 and 4.
2. All occupations—for comparisons of number of deaths presented in table 2 and online supplemental table S1.
3. All professional occupations—for comparisons of mortality risk presented in online supplemental tables S2 and S3.

Professional occupations are those which are classified by ONS as major occupational group 2, these are occupations which require a degree or equivalent period of relevant work experience and include, but are not limited to: scientists, engineers, architects, doctors, nurses, radiographers, physiotherapists, social workers and solicitors (for further information, see reference 11).

### Outcome definition
ONS defines deaths with COVID-19 as those where COVID-19 or suspected COVID-19 was mentioned anywhere on the death certificate, including in combination with other health conditions. If a death certificate mentions COVID-19, it will not always be the main cause of death but may be a contributory factor.

### Statistical analysis
We calculated the relative risk (RR) of death for educational professionals, lunchtime and teaching assistants compared with all working-aged (20–64 years old) people and compared teachers with all professionals in England and Wales stratified by sex, for this we used published ONS data on total deaths and age-adjusted mortality rates per 100 000 for each group. We derived the denominator (total population) for each group of interest as:

*Risk=mortality rate per 100 000 taken from ONS data/100 000.*

We calculated 95% CIs and p values using the formula:

$CI = exp(ln(RR) \pm 1.96 \times \sqrt{(1/deaths + 1/control\ deaths - 1/population - 1/control\ population)}$

$P = exp((-0.717 \times (RR/SE)) - (0.416 \times (RR/SE)^2))$

The ONS data included deaths for eight subgroups of educational professionals; we included those with a sufficient number of COVID-19 deaths (≥10) in our analysis. In addition, sufficient data were available to include female teaching and lunchtime assistants in this analysis. We also performed a fixed-effect meta-analysis across the four occupational groups working in schools among women.

We estimated mortality rates for all other causes (other than COVID-19) by calculating the difference between all-cause mortality and mortality with COVID-19 for each minor occupational group as defined by ONS.[9 11]

**Table 2** Number of deaths among working-aged (20–64 years) people who were working in education in England and Wales, 9 March–28 December 2020

| | Men | | | | | | Women | | | | | |
|---|---|---|---|---|---|---|---|---|---|---|---|---|
| Description | Deaths involving COVID-19 | All causes of death | All causes 5-year average | Excess | % excess | Excess due to COVID-19 | Deaths involving COVID-19 | All causes of death | All causes 5-year average | Excess | % excess | Excess due to COVID-19 |
| Secondary education teaching professionals | 29 | 241 | 233 | 8 | 3 | 100% | 23 | 156 | 173 | −17 | −10 | NA |
| Primary and nursery education teaching professionals | 4 | 31 | 30 | 1 | 3 | 100% | 19 | 327 | 368 | −41 | −11 | NA |
| Special needs education teaching professionals | 1 | 12 | 8 | 4 | 50 | 25% | 3 | 27 | 33 | −6 | −18 | NA |
| Teaching and other educational professionals- not elsewhere defined (n.e.c) | 8 | 56 | 32 | 14 | 44 | 57% | 7 | 70 | 61 | 9 | 15 | 78% |
| School secretaries | 0 | 3 | 2 | 1 | 50 | 0% | 4 | 58 | 53 | 5 | 9 | 80% |
| Teaching assistants | 5 | 28 | 19 | 9 | 47 | 56% | 37 | 396 | 307 | 89 | 29 | 42% |
| Educational support assistants | 1 | 7 | 8 | −1 | −13 | NA | 3 | 55 | 58 | −3 | −5 | NA |
| School midday and crossing patrol occupations | 2 | 11 | 5 | 6 | 120 | 33% | 18 | 169 | 149 | 20 | 13 | 90% |
| All those working in schools | 50 | 389 | 337 | 52 | 15 | 96% | 114 | 1258 | 1202 | 56 | 5 | 100% |
| All professional occupations | 419 | 3144 | 2765 | 379 | 14 | 100% | 279 | 2696 | 2477 | 219 | 9 | 100% |
| All occupations | 4225 | 33904 | 29745 | 4159 | 14 | 100% | 1742 | 18419 | 16528 | 1891 | 11 | 92% |

Deaths among working-aged (20–64 years) people who work in education in England and Wales, 9 March–28 December 2020. This table shows the number of deaths by occupation in 2020 and the average for the previous 5 years among those working in schools aged 20–64 years, also includes figures for all occupations and all professionals for comparison. The number of excess deaths was calculated by subtracting the 5-year average number of deaths (provided by ONS) from the number of deaths from all causes between 9 March and 28 December 2020 for each occupation and occupational group. The number of excess deaths was converted into a percentage of the 5-year average (percentage excess). The percentage excess explained by COVID-19 was calculated by dividing the number of COVID-19 deaths by the number of excess deaths and multiplying this by 100. We set the maximum percentage excess deaths to be 100.
ONS, Office for National Statistics.

**Table 3** COVID-19 deaths for educational professionals aged 20–64 years compared with all working-aged adults, between 9 March and 28 December 2020

| Description | Men | | | | | Women | | | | |
|---|---|---|---|---|---|---|---|---|---|---|
| | Risk* | Risk for all workers | RR | 95% CI | P value | Risk* | Risk for all workers | RR | 95% CI | P value |
| All educational professionals | 0.00018 | 0.00031 | 0.59 | 0.46 to 0.75 | 0.000018 | 0.000098 | 0.00017 | 0.58 | 0.46 to 0.73 | 0.0000058 |
| Higher education teaching professionals | 0.00012 | 0.00031 | 0.37 | 0.20 to 0.68 | 0.0016 | NA† | | | | |
| Further education teaching professionals | 0.00025 | 0.00031 | 0.79 | 0.42 to 1.46 | 0.46 | NA† | | | | |
| Secondary education teaching professionals | 0.00039 | 0.00031 | 1.25 | 0.87 to 1.80 | 0.23 | 0.00021 | 0.00017 | 1.26 | 0.84 to 1.90 | 0.27 |
| Primary and nursery teaching professionals | NA† | | | | | 0.00010 | 0.00017 | 0.60 | 0.38 to 0.93 | 0.024 |
| Teaching assistants | NA† | | | | | 0.00015 | 0.00017 | 0.89 | 0.65 to 1.23 | 0.50 |
| Lunchtime assistants | NA† | | | | | 0.00019 | 0.00017 | 1.14 | 0.72 to 1.82 | 0.58 |

RRs for death occurring with COVID-19 for education professionals and school staff aged 20–64 years compared with the total working-aged population, calculated using age-adjusted mortality rates by occupation between 9 March and 28 December 2020 from the ONS dataset.
*Total deaths divided by denominator.
†Less than 10 deaths, rate not reported.
ONS, Office for National Statistics; RR, relative risk.

We produced scatterplots of 'all-cause' versus 'with COVID-19' mortality and 'all other causes' versus 'with COVID-19' mortality by occupational group for men and women separately and calculated Pearson's correlation coefficient for these plots.

We also calculated proportionate mortality ratios as mortality rates from COVID-19 divided by all-cause mortality for occupations working in schools.

Finally, we used our estimates from above for the SE and the denominators for each occupational group to estimate the variance for COVID-19 mortality. We then performed a weighted least squares regression analysis (weighted by 1/variance) of COVID-19 mortality against other cause to determine the increase in COVID-19 for one death increase in other cause mortality across occupational groups.

### Patient and public involvement
The public were not involved in the design or conduct of this study.

### RESULTS
Death rates from COVID-19 and all causes among all educational professionals are presented alongside death rates for all professionals and all working-aged people by time period in table 1.

Between 8 March 2020 and 28 December 2020, there were 68 757 deaths from all causes among the working-aged (20–64 years) population in England and Wales, 7961 (12%) of these deaths involved COVID-19, of which, 2494 had occurred prior to 20 April, 2267 occurred between 20 April and 25 May, and 3200 occurred between 25 May and 28 December 2020.

There were 1326 deaths from all causes among educational professionals during this period; 139 of those deaths were thought to involve COVID-19.

COVID-19 was involved in one in eight male deaths and one in nine female deaths among working-aged people; among education professionals, these figures were approximately 1 in 8 deaths among men and 1 in 11 among women. COVID-19 was involved in a higher proportion of deaths early on in the pandemic; data covering the 6 weeks of the first peak in the UK showed that COVID-19 was involved in one in four deaths in male and one in six deaths in female educational professionals during this time. During the 10 months covered by the ONS data, all-cause and COVID-19 mortality rates among educational professionals appeared to be lower than for all working-aged adults and similar to those for all professionals.

**Table 4** All-cause mortality for school staff aged 20–64 years compared with all working-aged adults, between 9 March and 28 December 2020

| Description | Men | | | | | Women | | | | |
| --- | --- | --- | --- | --- | --- | --- | --- | --- | --- | --- |
| | Risk* | Risk for all workers | RR | 95% CI | P value | Risk* | Risk for all workers | RR | 95% CI | P value |
| All educational professionals | 0.0015 | 0.0026 | 0.60 | 0.55 to 0.65 | $1.74 \times 10^{-29}$ | 0.0011 | 0.0016 | 0.69 | 0.65 to 0.74 | $6.14 \times 10^{-23}$ |
| Higher education teaching professionals | 0.00095 | 0.0026 | 0.37 | 0.30 to 0.47 | $2.00 \times 10^{-16}$ | 0.00073 | 0.0016 | 0.46 | 0.35 to 0.61 | $1.13 \times 10^{-07}$ |
| Further education teaching professionals | 0.0017 | 0.0026 | 0.66 | 0.52 to 0.84 | 0.00077 | 0.00096 | 0.0016 | 0.61 | 0.47 to 0.79 | 0.00015 |
| Secondary education teaching professionals | 0.0032 | 0.0026 | 1.26 | 1.11 to 1.43 | 0.00033 | 0.0015 | 0.0016 | 0.92 | 0.78 to 1.07 | 0.28 |
| Primary and nursery teaching professionals | 0.0024 | 0.0026 | 0.94 | 0.66 to 1.33 | 0.72 | 0.0019 | 0.0016 | 1.18 | 1.06 to 1.32 | 0.0023 |
| Teaching assistants | 0.0017 | 0.0026 | 0.65 | 0.45 to 0.95 | 0.024 | 0.0017 | 0.0016 | 1.05 | 0.95 to 1.16 | 0.3 |
| Lunchtime assistants | NA† | | | | | 0.0018 | 0.0016 | 1.14 | 0.98 to 1.33 | 0.08 |

RRs for death occurring from all causes for education professionals and school staff aged 20–64 years compared with the total working-aged population, calculated using age-adjusted mortality rates by occupation between 9 March and 28 December 2020 from the ONS dataset.
*Total deaths divided by denominator.
†Less than 10 deaths, rate not reported.
ONS, Office for National Statistics; RR, relative risk.

### Number of deaths by individual occupations and combined across all school workers compared with 5-year average rates

We examined number of deaths by occupation among those working in schools aged 20–64 years (table 2) and among those aged 65 years or older (online supplemental table S1). Among working-aged people when deaths across all occupations working in schools were combined, we observed 15% more deaths (excess) compared with the 2015–2019 average for the same period among men and 5% more among women. For women, this was less than the excess observed across all occupations and among all professionals; for men, it was similar. Deaths with COVID-19 appeared to account for almost all the excess among men, whereas among women there were twice as many deaths with COVID-19 among school workers as there were excess deaths.

Among the older age group, there were much higher numbers of deaths among those working in schools compared with their 5-year average (74% more deaths among men and 37% among women). However, only 33% of excess deaths in men and 37% in women were thought to involve COVID-19. The group of all professionals and of all occupations had fewer excess deaths as a proportion of their 5-year average but a higher proportion of the deaths involved COVID-19 compared with those working in schools.

### Comparison of COVID-19 and all-cause deaths among educational professionals and those working in schools compared with the working-aged population

Table 3 shows that compared with the total working-aged population, there was strong evidence that the risk of dying with COVID-19 was lower among all educational professionals combined ($RR_{males}$ 0.59, 95% CI 0.46 to 0.75, $RR_{females}$ 0.58, 95% CI 0.46 to 0.73). There was also some evidence of a lower risk of death with COVID-19 among female primary and nursery educational professionals (RR 0.60, 95% CI 0.38 to 0.93). For both male and female secondary school teachers, the RR estimate shows a 25% higher risk of death from COVID-19, although there was a great deal of uncertainty around these estimates ($RR_{males}$ 1.25, 95% CI 0.87 to 1.80, $RR_{females}$ 1.26, 95% CI 0.84 to 1.90). There was little evidence of a difference in risk between teaching assistants or lunchtime assistants compared with working-aged people. A fixed-effect meta-analysis across the four occupational groups working in schools in women showed that risk was similar to all working-aged people (RR 0.95, 95% CI 0.77 to 1.14), with weak evidence of heterogeneity across the groups ($I^2 = 55.2\%$, $X^2 = 6.697$, p=0.08).

Table 4 shows strong evidence that the risk of death from all causes during this pandemic period was lower for higher and further education teaching professionals compared with the working-aged population of England and Wales. The was also evidence that male secondary

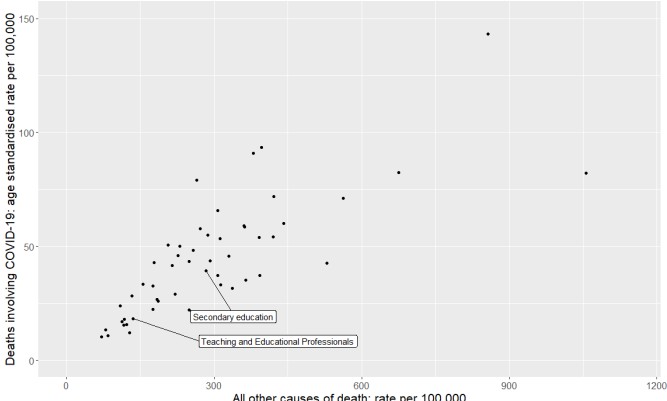

**Figure 1** Scatterplot of mortality rate (per 100 000) for COVID-19 and all other causes of death for men, using age-adjusted mortality rates between 9 March and 28 December 2020 from the Office for National Statistics. Unlabelled data points represent occupational groups in online supplemental table S4. Pearson's correlation coefficient=0.78, $p=1.4\times10^{-11}$.

school teachers (RR 1.26, 95% CI 1.11 to 1.43) and female primary school teachers (RR 1.18, 95% CI 1.06 to 1.32) had slightly higher mortality risk compared with the general population.

We also calculated RRs for mortality with COVID-19 and all-cause mortality in educational professionals versus all professionals (online supplemental tables S2 and S3). We found strong evidence that both male and female secondary school teachers had a higher risk of mortality with COVID-19 compared with all professionals. In addition, male and female primary and secondary school teachers had higher mortality risks for all causes compared with all professionals.

### Comparison of COVID-19 with other cause mortality across occupational groups

Figures 1 and 2 show that there are strong correlations between all other cause mortality and mortality with

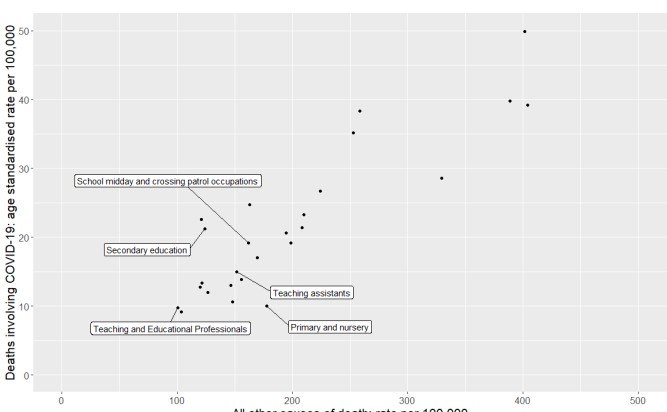

**Figure 2** Scatterplot of mortality rate (per 100 000) for COVID-19 and all other causes of death for women, using age-adjusted mortality rates between 9 March and 28 December 2020 from the Office for National Statistics. Unlabelled data points represent occupational groups in online supplemental table S5. Pearson's correlation coefficient=0.88, $p=1.7\times10^{-09}$.

COVID-19. Weighted least squares regression analysis showed that for every death from other causes, there were 0.11 (95% CI=0.09 to 0.13, $p=2.0\times10^{-16}$) deaths from COVID-19, with no meaningful difference between men (0.11, 95% CI 0.09 to 0.13, $p=2.49\times10^{-12}$) and women (0.10, 95% CI 0.08 to 0.13, $p=4.41\times10^{-9}$).

We also plotted mortality with COVID-19 versus all-cause mortality on separate scatterplots (online supplemental figures S1 and S2) and found similar results.

### Ratio of mortality with COVID-19 to other causes

Among men, the ratio of deaths with COVID-19 to other causes of death, ranged from 1:3 among male nurses and midwifes to 1:13 among male elementary construction occupations (online supplemental table S4). The ratio of COVID-19 to other causes of death for male secondary school teachers was 1:7, which was the same as the average for all male professionals and all working-aged men. Among women (online supplemental table S5), the ratio of mortality with COVID-19 to other causes of death ranged from 1:5 among welfare professionals (social workers and probation officers) to 1:18 among primary and nursery school teachers; ratios for female secondary school teachers, teaching assistants and lunchtime assistants were 1:6, 1:10 and 1:8, respectively. Proportionate mortality rates (COVID-19 deaths divided by deaths from all causes) are presented in online supplemental table S6.

### DISCUSSION
### COVID-19 deaths among educational professionals and others working in schools

We used routinely collected data on mortality for the ONS in the UK to study deaths with COVID-19 among teachers and other school staff, during the pandemic in 2020. We found that all educational professionals combined had fewer deaths per 100 000 occurring with COVID-19 compared with the general population. This is to be expected since the total working-aged population of England and Wales will include those who are unable to work due to ill health. The group 'educational professionals' includes those working in universities and school inspectors, many of whom have been working from home throughout the pandemic. We therefore investigated excess deaths specifically among school teachers and other staff working in schools and compared them with all those currently working and all professionals. There were fewer deaths than the 5-year average among female primary and secondary school teachers and the number of deaths among male teachers was similar to the 5-year average. There were more deaths among teaching assistants compared with the 5-year average, but only around half of the excess deaths were thought to involve COVID-19. For all women working in schools combined, there was a small increase in the number of deaths (5%) compared with the 5-year average, but the number of deaths occurring with COVID-19 was greater than the excess, suggesting that some of those dying with

COVID-19 may have died over the course of a normal year from other causes. For those working in schools who were aged 65 years and over, there were large excesses in deaths compared with the average for the previous 5 years (74% for men and 37% for women), but only around one-third of the excess deaths were thought to involve COVID-19. It is possible that some excess deaths were due to undiagnosed COVID-19 among those who did not have classic symptoms,[13] but this is less likely for later deaths as the amount of testing for COVID-19 increased dramatically over time in the UK. We do not have information on other causes of death within this population over this time period, however in 2019, the most common causes of death among over 65s in England and Wales were heart disease and cancer.[14] It is possible that more school staff were dying from other causes during this period due to delayed treatments for other conditions or due to an unwillingness to seek help for fear of contracting COVID-19 or of overburdening the healthcare system. It is also possible the number of school staff in this group has increased over the last 5 years. Further research is needed to determine whether these represent numerator–denominator bias or if there are true excesses and if so the cause of these excess deaths.

### RR of death with COVID-19 among school staff

We found that secondary school teachers had around a 25% higher risk (in both men and women) of dying with COVID-19 compared with the general population, although the evidence was weak. There was stronger evidence of an increased risk to secondary school teachers compared with all professionals. However, all-cause mortality was also increased among secondary school teachers compared with all professionals.

Proportionate mortality ratios (mortality from a specific cause versus 'all-cause' or 'all other cause mortality') are routinely used in occupational epidemiology in situations where calculation and adequate standardisation of mortality rate ratios is not possible.[15] The rationale for doing this is that if an occupation has a high risk for a specific disease, the proportion of deaths due to that disease will be increased relative to deaths from other causes in the group. We therefore plotted the correlation between age-adjusted mortality for all causes and mortality with COVID-19 across minor occupational groups for whom risk was reported in the ONS dataset, the two measures are bound to be correlated to a certain extent because all-cause mortality includes with COVID-19. However, we also found a strong positive correlation between mortality with COVID-19 and non-COVID-19. This is consistent with findings from others which show that deaths from COVID-19 closely match 'normal' risk within the UK population.[13 16] Mortality from any cause among all educational professionals and teachers was towards the lower end of the distribution. For male secondary teachers, female teaching and lunchtime assistants, the ratio of mortality with COVID-19 to all-cause mortality was consistent with other occupational groups;

female primary school teachers and female secondary school teachers appeared to be outliers in opposite directions, however the number of deaths in these groups was small and a meta-analysis across occupations working in school showed only weak evidence of heterogeneity.

### Consistency with other studies

The data we examined from ONS are consistent with the Public Health Scotland study[2] and with the Swedish study[7] which both showed similar or lower risk of hospitalisations due to COVID-19 among teachers compared with other occupations, but our findings are also compatible with a slightly higher risk of COVID-19 mortality among secondary school teachers. Mutambudzi and colleagues[17] investigated the risk of hospitalisation or death by occupation after testing positive for COVID-19 among 120 075 working participants (aged 49–64 years) in the UK Biobank study; they found weak evidence of more hospitalisations or deaths with COVID-19 among education workers compared with non-essential workers up to July 2020 (OR 1.59, 95% CI 0.87 to 2.91) after adjusting for potential confounders. However, UK Biobank is a highly selected population, which is not representative of the UK population due to a response rate of just 5%.[18]

A study of data on teacher absence due to COVID-19 in England during the autumn term (September–December 2020) found that the proportion of teachers absent due to infection appeared to be similar in both primary and secondary schools.[19] The ONS schools infection survey found that primary and secondary staff had similar antibody positivity rates for SARS-CoV-2 (15% and 16%, respectively) at the end of autumn term, which reflected exposure during the time when in-person teaching was taking place; positivity rates among all teachers were slightly lower than among all working-aged adults (18%).[20] Despite this, primary school teachers in England and Wales had a lower risk of death with COVID-19 compared with secondary school teachers (although CIs were overlapping).

Very few studies have examined risk among teaching assistants and lunchtime assistants as separate groups; we found that despite more deaths occurring in this group in 2020 compared with the previous 5 years, they were not at increased risk of death due to COVID-19 or any cause compared with working-aged people. A study carried out in California used modelling to compare deaths from all causes during the pandemic and found an OR for teaching assistants of 1.28, but no CIs were provided for that study.[21]

### Strengths and limitations of the study

We used routinely collected data on mortality which include all deaths in England and Wales, and results are therefore unlikely to be due to ascertainment bias and will be representative of the working-aged population of these countries. However, we did not have access to individual-level mortality data so were not able to account for potential confounders such as comorbidities or household

size. For our RR calculations, our comparison group also included school staff (although they only made up a very small proportion of the total) because we did not have age-adjusted mortality rates excluding teachers. The age-adjusted mortality rates calculated by ONS were based on denominators taken from a survey conducted in 2019. If the school workforce has changed as a result of the pandemic, these rates will be inaccurate; if any changes in the school workforce are systematically different to other professions, the results will be biased. While the size of the school workforce in England remained fairly stable between 2015 and 2019, there was a marked decrease in the number of teachers retiring (from 17 853 teachers retiring in 2015 to 12 062 teachers in 2019),[22] suggesting that the average age of teachers may have increased over this time period. We do not know what impact the pandemic has had on the size and age distribution of those working in schools, it could be that more vulnerable older staff have retired or conversely staff may have stayed on to help with staff shortages due to teachers having to isolate. Therefore, we were unable to adequately assess the extent to which risk of death had changed during the pandemic period for these occupational groups especially among those aged over 65 years. In addition, the number of deaths was extremely small and subject to random fluctuations in some groups. While our results are likely to be generalisable to countries which are similar to the UK, they may not be generalisable to developing countries and countries with different social structures.

In the period covered by these data, many school staff worked remotely for large periods of time during 2020. A survey carried out by the global education company Tes found that just 22% of teachers were engaged in face-to-face teaching during the first lockdown.[23] The more vulnerable staff are likely to have worked from home even when schools were fully open, which will have an impact on their workplace exposure. However, all teachers were potentially exposed to COVID-19 up until schools closed on 20 March 2020 and deaths up to 25 May, just 2 months after schools were closed, accounted for approximately two-thirds of all deaths occurring over the study period (table 1). Given the lag between infection and death, many people dying up to 25 May could have been infected prior to school closures. We do not have data on deaths by date and occupation; however, out of a total of 68 186 deaths which occurred due to COVID-19 in England and Wales between the dates covered by these data,[24] only 5018 deaths (7.4%) occurred between 25 May 2020 and 30 September 2020, a period during which those dying were unlikely to have been infected in school due to school closures. Therefore, the vast majority of people dying of COVID-19 during the period covered by the study could have been infected when schools were open.

## CONCLUSION

Teachers, teaching and lunchtime assistants aged 20–64 years were not at high risk of death relative to the working-aged population in England and Wales during the COVID-19 pandemic. For occupations working in schools, COVID-19 mortality was generally proportionate to all-cause mortality. Female primary school teachers and female secondary school teachers seemed to be outliers in opposite directions; however, evidence for heterogeneity in COVID-19 mortality risk among school workers was weak. For policymakers who are considering the impact of schools being open in future SARs-CoV2 pandemics, it will be important to note that staff were not at high risk of death compared with other occupations. Although we did not address these in this manuscript, other outcomes such as hospitalisation due to COVID-19 and long COVID-19 are also important considerations. In addition, for COVID-19, the emergence of more infectious strains since the data for this study were collected plus vaccination of school staff are likely to have changed the situation in schools.

**Author affiliations**
¹Population Health Sciences, University of Bristol, Bristol, UK
²MRC- Integrative Epidemiology Unit, University of Bristol, Bristol, UK
³Experimental Psychology, University of Bristol, Bristol, UK

**Contributors** SJL and GDS conceived the idea for the manuscript and designed the analyses. SJL and KD performed the analyses. SJL, GDS, CLR and MRM all contributed to the interpretation of the results. All authors contributed to drafting the manuscript and have seen and approved the final version. SJL is the guarantor for this work and accepts full responsibility for the work.

**Funding** SJL, CLR and GDS have received a grant from the National Institute for Health Research (NIHR) and UK Research and Innovation (UKRI) COVID-19 mapping and mitigation in schools (CoMMinS). GDS is director of the MRC Integrative Epidemiology unit and receives support from the MRC (MC_UU_00011/1), CLR and MRM are members of the MRC Integrative Epidemiology Unit and also receive support from the MRC (MC_UU_00011/5) and (MC_UU_00011/7).

**Competing interests** None declared.

**Patient consent for publication** Not required.

**Ethics approval** Ethical approval was not sought for this project because we used publicly accessible summary statistics of a national mandatory data collection.

**Provenance and peer review** Not commissioned; externally peer reviewed.

**Data availability statement** Data are available in a public, open access repository.

**ORCID iDs**
Sarah J Lewis http://orcid.org/0000-0003-4311-6890
George Davey Smith http://orcid.org/0000-0002-1407-8314

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
