## [Reviewer comments · BMJ Open]

ARTICLE DETAILS

TITLE (PROVISIONAL)	Was the risk of death among the population of teachers and other school workers in England and Wales due to Covid19 and all causes higher than other occupations during the pandemic in 2020? An ecological study using routinely collected data on deaths from the Office for National Statistics
AUTHORS	Lewis, Sarah; Dack, Kyle; Relton, Caroline; Munafò, Marcus; Davey Smith, George

VERSION 1 – REVIEW

REVIEWER	Almagor, J University of Glasgow, MRC/CSO Social and Public Health Sciences Unit
REVIEW RETURNED	10-May-2021

GENERAL COMMENTS	The researchers evaluated the relative risk of death from Covid19 for school teachers and others working in schools compared to other occupation groups. In the introduction they suggest that: “Whether or not teachers and others working in schools are at higher risk from Covid19 as a result of schools being open is central to decisions on school reopening, but until recently there has not been good data on this.” (p.1 row 34) They reiterate the lack of evidence by stating that: “There is an absence of evidence on the risk to teaching and lunchtime assistants working in schools.” (p.4 row 7). These statements suggest that the exposure of teachers to the school settings (interactions with multiple children in classes and with other staff members) may increase their risk to contract Covid19, which may lead to increased death rates from Covid19. The researchers calculated the relative risk of dying with Covid19 for teachers using several risk indices. Several difficulties arise with the risk calculation: The researchers used mortality data for the period 8 March - 28 December 2020. During this period schools were fully opened only from September to 11 December (as noted in p.1 row 27) or for 3 out of 9 months, and only about 20% of the teachers worked during the full period of March-December teaching children of critical workers and vulnerable children. Therefore, the risk is calculated for a period in which most teachers (about 80%) were mostly not working in the school settings, not directly exposed to children; and for a relatively smaller group of teachers (20%) that were exposed to school settings for a longer time period. As the risk is calculated for the
---

	whole teacher population together an elevated risk for the teachers who worked in schools for most of the period will be averaged over the whole teacher population and consequently reduced. My point is that the data the researchers use is limited in identifying the actual risk of the school setting. I think the research question should be more explicit, stating whether the risk refers to the school settings or just broadly to the population of teachers and other supportive staff during the first year of the pandemic with no consideration to whether they actually worked in the school settings. I suggest the researchers describe the findings regarding the risk in an appropriate context. They should add to the discussion a section in which they interpret what the relative risk they calculated includes. To what extent it answers what seems to be the research question: what is the risk of dying from Covid19 for teachers and others working in the school settings relative to other professions? Does the risk they calculated actually reflect the risk to teachers who worked most of the time from home? What can policy makers learn from the findings regarding school opening during a pandemic? Can the researchers use the data to evaluate a differential risk to the group of teachers who mostly worked from home (80%) and for the other group who mostly worked from school (20%)? For example, they could use assumptions on the risk for the periods when teaching from home - maybe similar to the risk of the higher and further education teachers and extract the risk for the period when working in schools. Such a calculation could give an indication whether the actual risk for teaching in schools may be higher.
--	---

REVIEWER	Gordon, Aliza HealthCore Inc
REVIEW RETURNED	11-May-2021

GENERAL COMMENTS	General This paper analyzes Covid-19 and all-cause mortality in teachers and other school workers compared to the working aged population and other professionals. While the authors appear to have done much of what they could with the publicly available data they used, my main concern is that the paper does not fully address whether actually working in schools (rather than performing these jobs remotely) is associated with a higher risk of death from Covid-19 (or other causes) – which is the rationale for this study – since most of the teachers and other school workers were working remotely during at least half of the time period that was studied. This limitation is large, and I believe it greatly diminishes the usefulness of the findings; I'm not sure that this inherent flaw can be appropriately addressed. Additional comments and elaboration below. Methods  1. Second paragraph in methods should clearly lay out each type of analysis/comparison; should be in the same order that the results are presented. 2. The paper focuses on Mar-Dec deaths, even though schools were closed for most of this time. I assume it is not possible to
--

look at deaths just during the months that schools were open, plus an extra one or two for the lag between exposure to Covid-19 and death (March and April/May, then September through December). However, the earlier datasets with data through April and May may actually better approximate risk from in-person teaching. More attention should be given to these datasets, which are only covered to a small degree in Table 1, and not the rest of the paper.

Occupational exposure groups

3. How are “professionals” defined compared to other workers? (Provide example categories that would or would not be included as professional, or provide a definition.)

4. Page 5 line 3 – what group will be defined this way? Lunchtime assistants?

Statistical analysis

5. Page 5 lines 24 and 26 – unclear; if population was defined from mortality rate, shouldn't the risk just be derived from the mortality rate (divided by 100,000)?

Results

6. Organization of results is confusing. Table 1 is very similar to Tables 2 and 3 (taking the numbers from Table 1 and turning them into relative risks for Tables 2 and 3). There is subheader of results for Tables 2 and 3 “Comparison of Covid19 and all-cause deaths among education professionals and those working in schools compared to the working aged population”, but the text for the results regarding Table 1 already compare these groups (page 6 line 17: “Table 1 shows that during the 10 months covered by the ONS data all-cause and Covid19 mortality rates among educational professionals were lower than for all working aged adults and similar to those for all professionals.”). But Tables S1 and S2, which are presented before Tables 2 and 3, compare 2020 to prior years, which is a different method with different data; this should be its own separate section (and maybe not just supplementary tables).

Strength and limitations of the study

7. Most limitations were addressed appropriately. However, I think the largest limitation is that this study combines “occupation risk” when teachers/school workers were in schools and when they were working remotely. If the main rationale of the study is “Whether or not teachers and others working in schools are at higher risk from Covid19 as a result of schools being open is central to decisions on school reopening,” (page 3 line 34) I am not convinced that the study addresses this issue. The study finds that teachers and other school workers (with the exception of secondary school teachers) were not at higher risk of death from Covid19 compared to other occupations, but this cannot be assumed if school had been in-person throughout the course of the study, since the majority of the time schools were closed. This is addressed briefly at the very end of the limitations section, but I think it is a bigger issue than how it is described, and casts doubt on the significance of the findings of the study. While this may be impossible to address from the available data, one suggestion that may help is given above (Methods comment #2).

Figures 1 and 2

	8. What do unlabeled data points represent? The other occupational categories listed in Tables S5 and S6? This needs to be clarified/referred to (if it is the same list). 9. How much do the teaching/school worker data points deviate from the general trend line? Outliers are discussed, but it is not stated how these outliers were determined. From eyeballing the figure? Or was there a statistical method to determine that they were outliers? What is the magnitude of the difference? Other 10. When is “all workers” vs “total working age population” used? P8 line 5 – “all those currently working”; Tables 2 and 3 have columns for “all workers” – I assume these should all be changed to “working age population” (my understanding is there wasn’t data for “workers,” just “working aged”). 11. There are very many typos – incomplete sentences, missing words, etc. There are also many cases where sentences are hard to follow, not due to the content, but due to the writing. I felt that this was distracting while reviewing, and significant copy-editing is needed.
--	--

REVIEWER	Bachelet, Vivienne Universidad de Santiago de Chile, Facultad de Ciencias Médicas
REVIEW RETURNED	13-May-2021

GENERAL COMMENTS	As usual, I appreciate the opportunity to review for BMJ Open. I have read the manuscript titled “Risk of death among teachers in England and Wales during the COVID-19 pandemic”. Using publicly available data (death registrations from the Office for National Statistics), the authors “estimate occupational risk from Covid19 teachers...for England and Wales”. The study period was 20 March 2020, to 28 December, 2020 and the primary outcome was “death with Covid19”. The secondary outcome was “death from all causes”. My main comment regards the stated objective of the study. In other words, the study question. One might phrase the question as follows: What is the risk of dying from COVID-19 for teachers and others working in schools in England and Wales for the greater part of 2020? According to textbook definitions of ‘risk’, it generally refers to the probability of some untoward event occurring in the future. Assessing risk involves assessing risk factors—i.e., any measurable variable that might be causally associated with the occurrence of the event in the future. To elucidate whether a risk factor is in effect causally related to the event, prospective observational studies must be conducted comparing the exposed to the non exposed, which will give us risk estimations (relative risk, odds ratio). So, to answer this study question, a prospective observational study should have been designed to estimate the risk of dying from COVID-19 in teachers or those working in schools. Since confounding would have been a major issue in such a study, adjusted risk estimates would have had to be calculated after measuring many different potential confounding variables. But none of that was done in this study.
--

	So my assessment of this paper begins (and ends) by questioning the study design that was used: scant mortality data obtained from death registers broken down only by sex and occupation. Given the fact that I am at odds with the study design, the rest of the study follows a path of error, which explains why the reading is difficult. One is constantly trying to second guess why the authors did what and for what reason. The way to fix this problem is to reformulate the study question as follows, in my opinion: What are the death rates for teachers and people working in schools for COVID-19 and are they higher or lower than other professional groups or the working population? This study question calls for a descriptive study that could have been done using the data sources provided in this manuscript. Forget risk, forget p values, forget confidence intervals and any other analysis done in this paper. As it stands, this paper reports no results that we could work with to formulate policy or devise interventions, which is the reason why we research in the first place. If the paper were only to report rates, then we could use that to develop hypotheses of lower or greater rates among teachers and try to figure out what is the occupational context in which they carry out their job that is leading to the stated results. If the rates are lower, maybe they are remote-teaching; if the rates are higher, maybe schools are unprotected from viral dissemination within the school community. I do not recommend publishing this manuscript in its present form.
--	--

REVIEWER	Lega, Ilaria Istituto Superiore di Sanità, National Centre for Disease Prevention and Health Promotion
REVIEW RETURNED	24-May-2021

GENERAL COMMENTS	The issue of a possible increase of the risk of death among teachers during the Covid-19 pandemic is extremely interesting and has been widely debated. The paper is well written and documented and the authors are to be congratulated for their huge effort to provide timely evidence on the issue based on administrative data. However, I have a major concern related to the actual exposure of the subjects included in the study that prevent me from considering the paper suitable for publication in its current form. The authors claim that schools in England and Wales closed to most pupils on the 20th March 2020 and did not reopen fully until September 2020. Available evidence cited by the authors (e.g., the Swedish findings by Vlachos and colleagues) and major public health concerns are linked to SARS-CoV2 infection rate and COVID-19 mortality rate among teachers working in the schools. Nevertheless, the dataset the authors used for their analysis covered the period from 8th March to the 28th December 2020. School closure in England and Wales encompasses 5 months over 9 months of observation. In my view, the exposures of the
---

	subjects included in the study are likely to be mixed and diverse, and the related results could be misleading. The study has the potential to provide valuable information on the issue of the risk of death among teachers during the Covid-19 pandemic after a major revision, encompassing the choice of a more suitable period of observation focusing on school opening or on school closure.
--	--

VERSION 1 – AUTHOR RESPONSE

Reviewer: 1

I think the research question should be more explicit, stating whether the risk refers to the school settings or just broadly to the population of teachers and other supportive staff during the first year of the pandemic with no consideration to whether they actually worked in the school settings.

The title has now been changed to reflect that fact that we are considering deaths among the total population of school workers through-out the pandemic. The new title is “Was the risk of death among the population of teachers and other school workers in England and Wales due to Covid19 and all causes higher than other occupations during the pandemic in 2020? An ecological study using routinely collected data on deaths from the Office for National Statistics”

I suggest the researchers describe the findings regarding the risk in an appropriate context. They should add to the discussion a section in which they interpret what the relative risk they calculated includes. To what extent it answers what seems to be the research question: what is the risk of dying from Covid19 for teachers and others working in the school settings relative to other professions? Does the risk they calculated actually reflect the risk to teachers who worked most of the time from home? What can policy makers learn from the findings regarding school opening during a pandemic?

*This comment has been addressed by changing the title to reflect that we are presentating data on risk to the teaching and school worker population during the pandemic period in 2020. We have also extended the discussion on the risk of death given that teachers were working from home for most of the pandemic (see above). Finally we have added a sentence on what policy makers can learn from the findings in relation to schools being open during a pandemic to the end of our conclusion:
“For policy makers who are considering the impact of schools being open in future pandemics it will be important to note that staff were not at increased risk”.*

Can the researchers use the data to evaluate a differential risk to the group of teachers who mostly worked from home (80%) and for the other group who mostly worked from school (20%)? For example, they could use assumptions on the risk for the periods when teaching from home - maybe similar to the risk of the higher and further education teachers and extract the risk for the period when working in schools. Such a calculation could give an indication whether the actual risk for teaching in schools may be higher.

We thank the reviewer for this suggestion; unfortunately we do not have data on deaths by occupation and date of death, so we are unable to do this. However, as noted above the vast majority of deaths in England and Wales due to Covid19 (over 90%) occurred either when schools were open or within two months following school closures and we have expanded on this in the discussion.

Reviewer: 2

Second paragraph in methods should clearly lay out each type of analysis/comparison; should be in the same order that the results are presented.

This paragraph has now been changed to outline the different comparisons carried out in the order that they appear in the paper- “We used these data to: 1) describe the mortality rates among all educational professionals combined and present these alongside rates for all professionals and all working aged people for the time 3 different periods covered by this data (Table 1), 2) compare the number of deaths with historical data on deaths for the same occupational group over 5 years (2015 to 2019) for all school workers and for individual occupations (teachers, teaching assistants, school secretaries etc), which are presented alongside death rates among all occupations and all professional occupations over the same period of the pandemic (Table 2), 3) calculate the risk of Covid19 and all-cause mortality among teachers and other school staff compared with all working aged people (Table 3 and 4) and all professionals (Supplementary Table S3 and S4) and finally, 4) we investigated the ratio of mortality with Covid19 to mortality from other causes among different occupational groups to determine whether school staff were outliers (Supplementary Table S4-S6 and all Figures).

The paper focuses on Mar-Dec deaths, even though schools were closed for most of this time. I assume it is not possible to look at deaths just during the months that schools were open, plus an extra one or two for the lag between exposure to Covid-19 and death (March and April/May, then September through December). However, the earlier datasets with data through April and May may actually better approximate risk from in-person teaching. More attention should be given to these datasets, which are only covered to a small degree in Table 1, and not the rest of the paper.

We thank the reviewer for this suggestion; however, we have included all the deaths in our analysis, because only a small proportion (7.4%) of Covid19 deaths occurred in England and Wales between the end of May and the end of September, a period when those infected were unlikely to have been infected in schools, see above comment on this which has now been added to the discussion.

How are “professionals” defined compared to other workers? (Provide example categories that would or would not be included as professional, or provide a definition.)

The following has been added to the methods section – “Professional occupations are those which are classified by ONS as major occupational group 2. Professional occupations are those which require a degree or equivalent period of relevant work experience and include, but are not limited to; scientists, engineers, architects, doctors, nurses, radiographers, physiotherapists, social workers and solicitors, for further information see reference 11.”

Page 5 line 3 – what group will be defined this way? Lunchtime assistants?
Statistical analysis

Detailed information on occupational exposure groups used in this analysis is provided in the methods section under the subheading “occupational exposure groups”.

Page 5 lines 24 and 26 – unclear; if population was defined from mortality rate, shouldn't the risk just be derived from the mortality rate (divided by 100,000)?

We agree that this was unclear and unnecessary; risk has now been defined as

mortality rate per 100,000 taken from ONS data/100,000.

Organization of results is confusing. Table 1 is very similar to Tables 2 and 3 (taking the numbers from Table 1 and turning them into relative risks for Tables 2 and 3). There is subheader of results for Tables 2 and 3 “Comparison of Covid19 and all-cause deaths among education professionals and those working in schools compared to the working aged population”, but the text for the results regarding Table 1 already compare these groups (page 6 line 17: “Table 1 shows that during the 10 months covered by the ONS data all-cause and Covid19 mortality rates among educational professionals were lower than for all working aged adults and similar to those for all professionals.”).

We apologise for the confusion; however, Table 1 is distinct in that this describes mortality rates across all educational professionals by the 3 different time periods covered by the ONS datasets, whereas Tables 2 and 3 (now 3 and 4) present the relative risks of death from Covid19 and all causes across the whole period among individual occupations (i.e., secondary school teachers, teaching assistants) compared with all working aged people.

We have expanded the title of Table 1 to describe how this is distinct from the other tables- “Table 1- Age adjusted all cause and with Covid19 mortality rates per 100,000 population (95% confidence intervals) for all educational professionals, all professional occupations and all working aged adults in England and Wales by overlapping time periods covered by this data.”

We have also changed the text describing Table 1 slightly to reflect that this is a purely descriptive table. The sentence in question has been moved to the end of the description of Table 1 results and now reads- “During the 10 months covered by the ONS data all-cause and Covid19 mortality rates among educational professionals appeared to be lower than for all working aged adults and similar to those for all professionals”.

Tables S1 and S2, which are presented before Tables 2 and 3, compare 2020 to prior years, which is a different method with different data; this should be its own separate section (and maybe not just supplementary tables).

We have now added an addition subheading in our results section to distinguish the results displayed in Supplementary Table S1 (now Table 2) from those in Table 1- “Number of deaths by individual occupations and combined across all school workers compared with 5-year average rates”.

We have also moved Table S1 from the supplementary tables to be in the main manuscript. This is now Table 2.

What do unlabelled data points represent? The other occupational categories listed in Tables S5 and S6? This needs to be clarified/referred to (if it is the same list).

Unlabelled data points represent occupational groups in Supplementary Table S5 (now S4) and S6 (now S5). Text clarifying this has now been added to the figure legends.

How much do the teaching/school worker data points deviate from the general trend line? Outliers are discussed, but it is not stated how these outliers were determined. From eyeballing the figure? Or was there a statistical method to determine that they were outliers? What is the magnitude of the difference?

Table S6 shows proportionate mortality ratios with confidence intervals for teachers and other school workers and provides the ratio for all workers for comparison. We state in the discussion that “female primary school teachers and female secondary school teachers appeared to be outliers in opposite directions”, we did not carry-out a formal statistical test to determine this, but make this comment because female primary school teachers have a low proportionate mortality ratio and female secondary school teachers a high proportionate mortality ratio relative to all female workers.

When is “all workers” vs “total working age population” used? P8 line 5 – “all those currently working”; Tables 2 and 3 have columns for “all workers” – I assume these should all be changed to “working age population” (my understanding is there wasn’t data for “workers,” just “working aged”).

The total working age population is used in comparisons of mortality rates and risk because we had information on age-adjusted mortality rates for this group, and we did not have age-adjusted mortality rates across all occupations. However, for comparisons of the number of individuals dying among occupations working in school versus others we used data on number of deaths among all occupations as comparison as this is a more appropriate comparison group, due to a potential healthy worker effect. We have added the following to the occupational exposure section in the methods section:

“The comparison groups we used were:

*All working aged people – for comparisons of mortality rates and risk presented in Tables 1, 3 and 4.
All occupations – for comparisons of number of deaths presented in Tables 2 and S1.
All professional occupations – for comparisons of mortality risk presented in Tables S2 and S3.”*

There are very many typos – incomplete sentences, missing words, etc. There are also many cases where sentences are hard to follow, not due to the content, but due to the writing. I felt that this was distracting while reviewing, and significant copy-editing is needed.

The manuscript has been reviewed for typos.

Reviewer: 3

The way to fix this problem is to reformulate the study question as follows, in my opinion: What are the death rates for teachers and people working in schools for COVID-19 and are they higher or lower than other professional groups or the working population? This study question calls for a descriptive study that could have been done using the data sources provided in this manuscript. Forget risk, forget p values, forget confidence intervals and any other analysis done in this paper.

We have reworded the title of the manuscript to include the study question which is as the reviewer suggests- “Was the risk of death among the population of teachers and other school workers in England and Wales due to Covid19 and all causes higher than other occupations during the pandemic in 2020?”

We think it is important to include p-values and confidence intervals in the manuscript so that the reader can determine how much uncertainty there is in the data and where there is strong evidence of a difference between occupational groups.

As it stands, this paper reports no results that we could work with to formulate policy or devise interventions, which is the reason why we research in the first place. If the paper were only to report

rates, then we could use that to develop hypotheses of lower or greater rates among teachers and try to figure out what is the occupational context in which they carry out their job that is leading to the stated results. If the rates are lower, maybe they are remote-teaching; if the rates are higher, maybe schools are unprotected from viral dissemination within the school community.

We disagree with the statement that the paper reports no results that could help to formulate policy; we do report mortality rates as well as relative risks by occupation compared to other occupations, along with estimates of the uncertainty in the data (p-values and confidence intervals). This allows the reader to see where the rates or risks are higher among school staff compared to other occupations. We have added a statement on how this manuscript can inform policy at the end of the conclusion.

Reviewer: 4

The study has the potential to provide valuable information on the issue of the risk of death among teachers during the Covid-19 pandemic after a major revision, encompassing the choice of a more suitable period of observation focusing on school opening or on school closure.

We are unable to refine the period of study from the data we have. However, given that most deaths (93%) from Covid19 occurred either within 2 months of schools closing or when they were open think the dataset is useful to address our research question.

VERSION 2 – REVIEW

REVIEWER	Gordon, Aliza HealthCore Inc
REVIEW RETURNED	04-Aug-2021

GENERAL COMMENTS	Overall the clarity of the paper is much improved. I have two remaining comments: 1. I take your points about the timing of results and small percentage of deaths in England and Wales occurring between May and September. But we don't know if educational professionals had even lower rates of death during this time (e.g., maybe only 2% of educational profession deaths were May 25-Sept 25, higher than the total 7.4%). From the data presented in Table 1, we do see a slightly lower percentage of COVID deaths occurring in May25-Dec28 in educational professionals compared to others (and a higher percentage in March-April), particularly compared to all working age adults – see table below (derived from the data in Table 1) (each row sums to 100%). It is possible that if the last column was split to September and then December we would have seen even lower percentages for teachers in June-Sep and then higher in Oct-Dec, we just don't know. % of COVID deaths occurring in each period male Mar8-Apr20 Apr20-May25 May26-Dec28 female Mar8-Apr20 Apr20-May25 May26-Dec28
---

	WAA 32% 29% 39% 31% 27% 42% Prof 32% 34% 34% 33% 30% 38% Educ 36% 34% 30% 34% 28% 39% This would lead me to conclude that being at home may be affecting the numbers, and the main analysis of this paper should have a more restricted time period (either ending Apr20 or May25) to capture COVID deaths where exposure likely occurred while school was in session. I would prefer the earlier time period as the main analysis of the paper with full data through Dec as a sensitivity analysis, but the data through April (or perhaps May) should at least be analyzed as a sensitivity analysis for the primary analyses/tables to show that the comparable mortality rates to other populations were also seen in the early months where deaths were likely from cases occurring while school was open. 2. I agree with the importance of having a statement about what policymakers can take away from this research (as the authors state in the background “Whether or not teachers and others working in schools are at higher risk from Covid19 as a result of schools being open is central to decisions on school reopening”). However, I take issue with the new concluding sentence: “For policy makers who are considering the impact of schools being open in future pandemics it will be important to note that staff were not at increased risk” for the following reasons: a. Just because risk is similar to other occupations, it doesn’t mean it’s “not at increased risk” compared to school closures (which is the other option), particular since they are not being compared to other professionals working from home. They just may have average risk compared to a diverse group of people, some of whom are working from home and others who have professions requiring more social contact. b. Also, this is only about risk of death, not other risks like severe illness, hospitalization, or long COVID. (This is obviously the scope of the paper, but the concluding sentence should provide this framing.) c. These results may not be relevant to future pandemics, as children may be more likely to spread other diseases than the original and alpha variants of COVID. In fact the delta variant is likely more contagious in children than the original strain or variants occurring in 2020, so these results may not be even be generalizable to the evolving COVID situation with respect to transmission in schools. (Though of course vaccination changes the equation as well, and will obviously reduce the risk of death due to COVID among educational professionals, and this would be taken into account in any decisions about school openings.) The concluding thoughts should be updated to address these factors.
--	--

REVIEWER	Bachelet, Vivienne Universidad de Santiago de Chile, Facultad de Ciencias Médicas
REVIEW RETURNED	30-Jul-2021

GENERAL COMMENTS	Confidence intervals are inferential measures. We use inferential statistics when sampling a population when we have incomplete data, to estimate parameters of the population. Confidence intervals are warranted when there is incomplete data (samples) and, thus, uncertainty. The main purpose of confidence intervals is to indicate the precision or imprecision of sample estimates as population values. And yet, the authors use CI in Table 1 even when the table heading says explicitly “mortality rates...for ALL educational professionals...”. No sample. Likewise, using p-values in Table 3 should not be done. Hypothesis testing in this context is not warranted and is meaningless. If there are differences in mortality by occupation and exposure, we would be interested in the size of the difference for the measured outcome, not whether it is statistically significant.
---

VERSION 2 – AUTHOR RESPONSE

Reviewer 2 Comment 1-

I take your points about the timing of results and small percentage of deaths in England and Wales occurring between May and September. But we don't know if educational professionals had even lower rates of death during this time (e.g., maybe only 2% of educational profession deaths were May 25-Sept 25, higher than the total 7.4%). From the data presented in Table 1, we do see a slightly lower percentage of COVID deaths occurring in May25-Dec28 in educational professionals compared to others (and a higher percentage in March-April), particularly compared to all working age adults – see table below (derived from the data in Table 1) (each row sums to 100%). It is possible that if the last column was split to September and then December we would have seen even lower percentages for teachers in June-Sep and then higher in Oct-Dec, we just don't know.

% of COVID deaths occurring in each period

male Mar8-Apr20 Apr20-May25 May26-Dec28 female Mar8-Apr20 Apr20-May25 May26-Dec28

WAA 32% 29% 39% 31% 27% 42%

Prof 32% 34% 34% 33% 30% 38%

Educ 36% 34% 30% 34% 28% 39%

This would lead me to conclude that being at home may be affecting the numbers, and the main analysis of this paper should have a more restricted time period (either ending Apr20 or May25) to capture COVID deaths where exposure likely occurred while school was in session.

I would prefer the earlier time period as the main analysis of the paper with full data through Dec as a sensitivity analysis, but the data through April (or perhaps May) should at least be analyzed as a sensitivity analysis for the primary analyses/tables to show that the comparable mortality rates to other populations were also seen in the early months where deaths were likely from cases occurring while school was open.

Our response-

The Table the reviewer refers to Table 1 -Age adjusted all cause and with Covid19 mortality rates per 100,000 population (95% confidence intervals) for all educational professionals, all professional occupations and all working aged adults in England and Wales by overlapping time periods covered by this data. This is the only Table in the paper in which all education professionals are combined together in one group. Due to small numbers in specific occupational groups we do not have access to data on mortality by time period for secondary school teachers, teaching assistants etc. In this paper we aimed to investigate the impact of working in schools versus other environments on deaths from covid19, the group comprising 'education professionals' includes school inspectors and higher education professionals most of whom were working at home for the duration of the period covered. Therefore we conducted our analyses using data for specific occupational groups who work in schools across the whole period and are unable to repeat our analyses for the time period suggested. Even if we had approximate information on date of death for school workers and limited it to the periods suggested there would still be some ambiguity because; a) some people working in schools may have died from covid following a very long illness, so even those dying in July may have contracted covid19 in school in March, b) some teachers continued to work in schools even when they were closed to the majority of pupils, because they remained open for children of essential workers throughout the pandemic, c) school staff may have been working in school at the time of infection but have contracted covid19 elsewhere.

Reviewer 2 comment 2-

I agree with the importance of having a statement about what policymakers can take away from this research (as the authors state in the background "Whether or not teachers and others working in schools are at higher risk from Covid19 as a result of schools being open is central to decisions on school reopening"). However, I take issue with the new concluding sentence: "For policy makers who are considering the impact of schools being open in future pandemics it will be important to note that staff were not at increased risk" for the following reasons:

a. Just because risk is similar to other occupations, it doesn't mean it's "not at increased risk" compared to school closures (which is the other option), particular since they are not being compared to other professionals working from home. They just may have average risk compared to a diverse group of people, some of whom are working from home and others who have professions requiring more social contact.

b. Also, this is only about risk of death, not other risks like severe illness, hospitalization, or long COVID. (This is obviously the scope of the paper, but the concluding sentence should provide this framing.)

c. These results may not be relevant to future pandemics, as children may be more likely to spread other diseases than the original and alpha variants of COVID. In fact the delta variant is likely more contagious in children than the original strain or variants occurring in 2020, so these results may not be even be generalizable to the evolving COVID situation with respect to transmission in schools. (Though of course vaccination changes the equation as well, and will obviously reduce the risk of death due to COVID among educational professionals, and this would be taken into account in any decisions about school openings.)

The concluding thoughts should be updated to address these factors.

Our response-

We have taken the reviewers comments on board and revised the conclusion accordingly, this now reads:

“For policy makers who are considering the impact of schools being open in future pandemics it will be important to note that staff were not at high risk of death compared to other occupations. Although we did not address these in this manuscript, other outcomes such as hospitalisation due to Covid19 and long covid are also important considerations. In addition, for Covid19 the emergence of more infectious strains since the data for this study were collected plus vaccination of school staff are likely to have changed the situation in schools.”

Reviewer 3 comment –

Confidence intervals are inferential measures. We use inferential statistics when sampling a population when we have incomplete data, to estimate parameters of the population. Confidence intervals are warranted when there is incomplete data (samples) and, thus, uncertainty. The main purpose of confidence intervals is to indicate the precision or imprecision of sample estimates as population values. And yet, the authors use CI in Table 1 even when the table heading says explicitly “mortality rates...for ALL educational professionals...”. No sample.

Likewise, using p-values in Table 3 should not be done. Hypothesis testing in this context is not warranted and is meaningless. If there are differences in mortality by occupation and exposure, we would be interested in the size of the difference for the measured outcome, not whether it is statistically significant.

Our response-

Whilst it is true that the statistics used in this manuscript represent the whole population, the purpose of this paper is to compare mortality rates among teachers and other school staff to other occupations. In order to do this we have to take into account the stochastic nature of deaths over a given period of time within the population, particularly when the number of deaths being compared in each group is small. There will be random fluctuations in the number of people dying within each occupational group from year to year whether or not there is a pandemic and what we wanted to test was whether the differences we observed between occupational groups is greater than we would expect by chance given these fluctuations. For this reason we think it is appropriate to use confidence intervals and calculate p-values.

The following paragraph is taken from the Guidelines for Using Confidence Intervals for Public Health Assessment (wa.gov) Washington State Department of Health-

“There are a few in public health who believe that confidence intervals should not be used around estimates derived from ‘population’ statistics such as the death rate in a given population, because they believe there is no statistical uncertainty in such estimates. This belief is contrary to the statistical theory

underlying confidence intervals, and the biological and random processes governing the occurrence of events such as deaths and illnesses (Brillinger, 1986).” The paper referred to in the above quote is: Brillinger DR. The natural variability of vital rates and associated statistics [with discussion]. Biometrics 42:693-734, 1986.

VERSION 3 – REVIEW

REVIEWER	Gordon, Aliza HealthCore Inc
REVIEW RETURNED	01-Oct-2021

GENERAL COMMENTS	The paper has been much improved since the initial submission. While the paper provides information regarding how educational professionals fared with respect to mortality from COVID during March to December 2020, which is perhaps somewhat useful in its own right, I am still not convinced that the results are helpful in answering the question posed in the introduction: "Whether or not teachers and others working in schools are at higher risk from Covid19 as a result of schools being open is central to decisions on school reopening, but until recently there has not been good data on this" due to the timing and granularity of the data that were available to the authors.
--